## Research Article

climate change anxiety; psychological distress; mental health; hope; Arab countries

**Corresponding author:**
Souheil Hallit;
Email: souheilhallit@usek.edu.lb

S.H. and F.F.-R. are last co-authors.

# The direct and indirect effects of climate anxiety on psychological distress through hope: A multinational study in different Arab nations

Tigresse Boutros[1], Jana Abou Rjeily[1], Hanaa Ahmed Mohamed Shuwiekh[2], Mai Helmy[3], Kamel Jebreen[4,5], Abdallah Y. Naser[6], Mirna Fawaz[7,8], Inad Nawajah[9], Eqbal Radwan[10], Amer Abukhalaf[11], Diana Malaeb[12], Sahar Obeid[13], Muna Barakat[14,15], Souheil Hallit[1,16] and Feten Fekih-Romdhane[17,18]

[1]School of Medicine and Medical Sciences, Holy Spirit University of Kaslik, P.O. Box 446, Lebanon; [2]Department of Psychology, Fayoum University, Fayum, Egypt; [3]Psychology department, College of education, Sultan Qaboos University, Muscat, Oman; [4]Department of Mathematics, Palestine Technical University–Kadoorie, Hebron P766, State of Palestine; [5]Department of Mathematics, An-Najah National University, Nablus P400, State of Palestine; [6]Department of Applied Pharmaceutical Sciences and Clinical Pharmacy, faculty of pharmacy, Isra University, Amman, Jordan; [7]Faculty of Health Sciences, Nursing Department, Beirut Arab University, Beirut, Lebanon; [8]Rayak University Hospital, Bekaa, Lebanon; [9]Department of Mathematics, Hebron University, Hebron, Palestine; [10]Department of Biology, Faculty of Science, Islamic University of Gaza, Palestine; [11]Department of Construction and Real Estate Development, Clemson University, Clemson, United States; [12]College of Pharmacy, Gulf Medical University, Ajman, United Arab Emirates; [13]School of Arts and Sciences, Social and Education Sciences Department, Lebanese American University, Beirut, Lebanon; [14]Department of Clinical Pharmacy and Therapeutics, Faculty of Pharmacy, Applied Science Private University, Amman, Jordan; [15]Middle East University Research Unit, Middle East University, Amman, Jordan; [16]Department of Psychology, College of Humanities, Effat University, Jeddah 21478, Saudi Arabia; [17]The Tunisian Center of Early Intervention in Psychosis, Department of Psychiatry "Ibn Omrane", Razi hospital, Manouba, 2010 Tunisia and [18]Tunis El Manar University, Faculty of Medicine of Tunis, Tunis, 1007 Tunisia

## Abstract

The impact of climate change on mental health is becoming increasingly recognized. Previous studies on this subject have mainly assessed the direct and immediate emotional reactions to climate change anxiety, but the psychological aspects of this connection are yet to be investigated, especially in Arab societies. The current study aimed at investigating if hope can be a mediator in the relationship between climate change anxiety and psychological distress in Arab countries. A cross-sectional survey was conducted between February and June 2025 among 2,844 subjects from Egypt, Jordan, Palestine and Lebanon. The validated Arabic versions of the *climate change anxiety scale*, the *perceived Hope Scale* and the *patient health Questionnaire-4* were used for data collection. Hope was found to be a partial mediator in the relationship between climate change anxiety and psychological distress (indirect effect: $\beta = 0.003$; 95% CI [0.001, 0.005]). Higher levels of climate anxiety were associated with lower hope, which in turn was related to higher psychological distress. Climate change anxiety continued to be directly associated with psychological distress even after accounting for hope as a mediator. This study suggests that hope modestly and partially mediates the relationship between climate change anxiety and psychological distress. Therefore, and particularly in the Arab region, a multidisciplinary and collaborative approach aiming at reinforcing and strengthening hope may help with the mental health burden of climate-related anxiety.

## Impact statement

Our cross-sectional multinational study shows that hope can partially mediate the association between climate change anxiety and psychological distress. This mediating role of hope suggests that interventions aimed toward addressing climate-related distress may require approaches that extend beyond reducing anxiety alone. Although strengthening hope is unlikely to be sufficient on its own, it may complement broader mental health strategies such as screening for symptoms of anxiety and depression. In order to help patients manage their anxiety about climate change, collaborative approaches involving physicians, mental health practitioners, educators and policymakers, as well as collective assessment of hope and coping, are warranted, in both young and adult populations.

## Introduction

Climate change is increasingly recognized as a significant global health challenge affecting not only physical health but also psychological well-being (Cosh et al., 2024). Although the physical

impacts of climate change have been well studied, its emotional and psychological effects have only recently received careful scientific examination (Rocque et al., 2021). Climate change can influence mental health through three different mechanisms, including direct exposure to extreme weather events, indirect social and economic consequences and anticipatory awareness of environmental deterioration (Berry et al., 2010; Bourque and Cunsolo Willox, 2014). According to the Intergovernmental Panel on Climate Change (IPCC), climate-related events have significant effects on mental health globally, with these effects projected to worsen as temperature continues to rise (Pörtner et al., 2022; Ballew et al., 2024). Among these psychological outcomes, climate change anxiety (CCA) has received growing research interest (Cunsolo et al., 2020). It refers to persistent emotional distress and worry related to perceptions of climate change and its future consequences for the environment, societies and future generations (Clayton, 2020; Cruz and High, 2022). This psychological response has also been described as a distinct form of anxiety specifically related to the climate crisis (Pihkala, 2020b). Moreover, it is characterized by negative emotional responses due to experiences or anticipation of climate change-related events (Clayton and Karazsia, 2020). CCA is considered in the literature as one component of the broader construct of eco-anxiety, which is, according to the American Psychological Association, a "chronic fear of environmental doom" and refers to ecological crisis globally, comprising not only climate change but also other environmental problems like deforestation and pollution (Soutar and Wand, 2022; Asgarizadeh et al., 2023). According to research, CCA is not rare and is generally not considered a pathological condition but rather is regarded as an understandable and normal reaction to climate change (van Valkengoed, 2023). Although it is generally a reasonable response, it may lead to both adaptive and maladaptive outcomes. For instance, adaptive responses can include engagement in climate action, coping strategies and pro-environmental behaviors, whereas maladaptive outcomes can manifest as psychological ill-being and climate change denial (Tam et al., 2023). Recent studies indicate varying prevalence across different populations worldwide, influenced by demographic, geographic and environmental factors (Ahmed et al., 2025). In fact, the prevalence of clinically relevant CCA in a national survey conducted in Canada was 2.35% (Harper et al., 2025). However, in another study conducted among Palestinian undergraduate students, the prevalence of CCA was 6.1%, whereas in another one conducted in Iraq, the prevalence of CCA was 71.4% (Ahmead et al., 2025; Ahmed et al., 2025). A recent meta-analysis showed that certain groups may be more susceptible to CCA, including younger adults, women and personality traits such as neuroticism, which increase vulnerability (Kühner et al., 2025). In addition, CCA is increasingly considered a contributor to mental health and behavioral responses affecting diverse populations worldwide. Recent studies have found that CCA may be associated with worse insomnia, poorer mental health and general psychological distress (PD) and can impair everyday functioning and overall well-being (Harper et al., 2025; Ogunbode et al., 2025).

### The relationship between climate change anxiety and psychological distress

PD can be defined as a state of emotional suffering marked by symptoms of depression, such as sadness, hopelessness, anhedonia and anxiety such as tension, persistent worry and restlessness. It can often be accompanied by somatic complaints, including headaches,

fatigue and sleep disturbances, and is used in research as an indicator of mental health status (Drapeau et al., 2012). Exposure to climate-change-related events such as floods and wildfires has been associated with elevated risk of PD, anxiety, substance use, depression, post-traumatic stress disorder and even suicidality (Powell and Rao, 2023). Within the context of climate change, several studies suggest that higher levels of CCA are associated with increased levels of PD including symptoms of anxiety and depression across different populations (Lukacs et al., 2023; Reyes et al., 2023). For instance, in a study conducted among Filipinos, CCA explained 13.5% of the variance in overall mental health, suggesting a meaningful association with PD (Reyes et al., 2023). Furthermore, persistent CCA can interfere with routine activities and decision-making, thus increasing emotional strain, and can contribute to overall PD (Chan et al., 2024). For instance, in an Australian study involving 18,800 individuals, 38% of those who reported very high CCA experienced higher levels of PD that interfered with their daily lives compared to 22% of those who reported no concerns about climate change (Teo et al., 2024). In addition, research suggests that persistent CCA may increase susceptibility to PD by fostering feelings of hopelessness and perceived lack of control, which can therefore transform this reasonable anxiety into prolonged mental strain (Sangervo et al., 2022; Daeninck et al., 2023; Chan et al., 2025). Moreover, several mechanisms studied in the literature may mediate the impact of CCA on PD such as intolerance of uncertainty and future anxiety. For instance, a study conducted in Italy found that difficulties in coping with uncertainty and anxiety about the future may explain how CCA can exacerbate distress (Regnoli et al., 2024).

### Hope as a possible mediator in the relationship between climate change anxiety and psychological distress

Hope can be defined as both a cognitive concept and an emotion. In fact, hope has been frequently conceptualized as a cognitive concept grounded in Snyder's hope theory, in which pathways thinking refers to the ability to generate strategies in order to attain desired outcomes or to avoid undesired ones, coupled with agency thinking, which reflects the motivation to initiate and pursue these strategies to achieve goals (Snyder, 2000). On the other hand, hope as an emotional concept refers to an emotional state in which an individual recognizes that a situation could improve even if the outcome is uncertain, distinguishing it from optimism, which assumes a favorable outcome is certain (Frumkin, 2022). Although our study focuses on perceived hope, previous research has evaluated hope in the context of climate change. For instance, Ojala identified three ways in which individuals experience hope related to climate change, including strategies focused on meaning, reassessing the situation positively and relying on personal and collective action (Ojala, 2023). In our study, we chose to measure hope using the Perceived Hope Scale (PHS) because it is a short, reliable and culturally neutral instrument designed to measure perceived hope, reflecting at the same time both agency and pathways thinking (Fekih-Romdhane et al., 2025). Unlike the other context-specific measures such as the Climate Change Hope Scale (CCHS), which is not validated in Arabic, the PHS captures perceived hope as experienced by individuals in general, which may be a psychological resource for coping with climate-related distress (Pihkala, 2018).

Moreover, hope can either support adaptive responses and engagement when it is constructive or contribute to disengagement

and doubt when it denies the seriousness of climate change (Ojala, 2012). Therefore, hope in general may serve as an adaptive mechanism to manage stress and maintain emotional balance in individuals who are facing challenging situations such as climate change (Ilyas et al., 2025). Based on literature data, we suppose that hope may play the role of a possible mediator between CCA and PD. In fact, studies have shown that hope can influence how people experience CCA because as they tend to be more concerned and anxious about the future of the planet, hope may decrease, especially when they feel they have limited ability and action on their environment (Chan et al., 2025). Moreover, the literature indicates that hope may moderate the relationship between CCA and pro-environmental behavior (Leite et al., 2023). For instance, Ojala noted that when people feel powerless to influence collective solutions or see that environmental deterioration is inevitable, their sense of hope decreases, resulting in feelings of emotional exhaustion and resignation (Ojala, 2023). In addition, another study has found that CCA and hope often occur together, and more specifically that intense concern about climate change can limit the capacity for optimistic expectations and therefore limit climate action (Vercammen et al., 2023). In fact, persistent and intense anxiety about climate change can paralyze emotional responses, thus undermining hope and control (Gunasiri et al., 2022). This persistent increase in CCA may limit people's perception of hopeful outcomes, reducing their motivation to engage in proactive behavior and generating feelings of withdrawal and burnout, impairing their well-being (Daeninck et al., 2023).

On the other hand, hopelessness is not only a potential consequence of CCA, but it may also exacerbate PD. In fact, research has found that hope has been linked to better mental health whereas hopelessness has been associated with greater PD and symptoms of anxiety and depression (Frumkin, 2022). Moreover, hope has been associated with better coping, psychological resilience and overall well-being (Gallagher and Lopez, 2018). When facing climate change, lower levels of hope not only can result in disengagement, avoidance and burnout, but also can weaken a person's sense of meaning and expectation that improvement is possible (Vercammen et al., 2023). In addition, a systematic review has shown that hope is not only linked to greater subjective and social well-being, but it may also protect against persistent anxiety and help maintain a sense of control under uncertainty (Mortreux et al., 2025). Similarly, Powell and Rao noted that hope may be associated with lower levels of PD as it enables individuals to cope with feelings of uncertainty, powerlessness and despair (Powell and Rao, 2023). Taking all these findings into account, we propose that hope may serve as a possible mediator between CCA and PD.

### Rationale of the present study

The Arab region is highly susceptible to the negative effects of climate change. This vulnerability is intensified in the region due to more frequent natural disasters and extreme weather combined with political, social and economic conflicts, including rapid urbanization, rising poverty and population growth (UN-Habitat, 2022). As a consequence, such challenges not only threaten ecological systems but may also act as psychological stressors, increasing the risk of potential mental health problems in this region (Clayton and Crandon, 2025). For instance, an epidemiological study conducted among 18 Arab countries has found that climate change is associated with a range of mental health problems, including anxiety and depression (Arnout, 2023). These results indicate that climate change is not only an ecological and socioeconomic challenge in

the Arab region but also can be a source of psychological strain. Among the psychological consequences of climate change, CCA has been reported in different Arab populations with countries such as Jordan, Lebanon, Palestine and Egypt exhibiting moderate yet meaningful levels of CCA (Arnout, 2023). Although moderate, CCA in these countries can still affect daily functioning and mental health, especially when coupled with socioeconomic and environmental stressors (Fernández et al., 2025; Qader et al., 2025). As a result, it is relevant to consider psychological resources that may help individuals cope with climate anxiety, such as hope. The need to examine hope in Arab countries cannot be attributed only to increased exposure to climate change. In fact, in environments where political instability, limited adaptive infrastructures and economic strain are common, individuals may perceive limited possibilities for effective governmental responses to climate change, which can increase feelings of lack of control and helplessness (Soutar and Wand, 2022). Hope may therefore serve as a psychological resource by managing PD and facilitating future-oriented thinking and a sense of personal agency despite limited external resources (Aldbyani et al., 2025). Therefore, understanding how hope functions in these contexts may guide culturally adapted mental health interventions, enabling policymakers to better address climate change distress. However, most evidence comes from Western contexts leaving the role of hope in CCA understudied in Arab countries (Betro', 2024; Rodriguez Quiroga et al., 2024; Mortreux et al., 2025). Taking into consideration the potential role of hope in buffering the psychological impacts of climate change, it is therefore important to examine how it functions in specific contexts, particularly in the Arab region where data remain scarce. Our study focuses mainly on four Arab countries: Jordan, Lebanon, Palestine and Egypt. These countries were selected in our study because they represent contexts of high vulnerability to climate change combined with political and socioeconomic stressors. In fact, they are constantly exposed to climate-related challenges such as water scarcity, rising temperatures and droughts, which create further pressure on public health and social systems (Endowment for International Peace, 2023). Therefore, these four countries form an appropriate regional context to examine the role of hope as a potential psychological resource for climate distress.

The present cross-country study sought to explore the potential role of hope as a psychological pathway between CCA and PD across four Arab countries. By examining whether hope can buffer PD related to CCA, our study addresses a gap in the region, where evidence on such a possible protective mechanism remains limited.

## Methods

### Ethical considerations

Our study protocol was approved by the Ethics and Research Committee of the Lebanese International University (reference number: 2022RC-051-LIUSOP). All methods in our study were performed in conformity with the relevant regulations and guidelines. In addition, each participant was required to give a written informed consent before submitting the online form.

### Study design and participants

This multinational cross-sectional study was carried out between February and June 2025 and included participants from four different Arab countries: Jordan, Lebanon, Palestine and Egypt. We used an online survey in order to recruit participants and relied on

the snowball sampling approach to maximize participation and ensure a diverse representation across national and demographic backgrounds. As a consequence, participants who met the inclusion criteria, provided informed consent and filled out the survey were encouraged to distribute the questionnaire to their close contacts and wider network.

The questionnaire was given out via a link on Google Forms that was shared across multiple social media platforms and also via email networks, which made it possible for a very wide geographic distribution. Before respondents were allowed to participate, each of them was presented with an introductory statement clearly explaining the goals of the study and the ground rules of confidentiality that we aimed for. The statement also disclosed to the participants that their participation was totally voluntary and that the answers would be kept anonymous. Participants were required to provide electronic informed consent by selecting the following confirmation statement before proceeding: "Yes, I acknowledge having read the above-mentioned information and I agree to participate in this study voluntarily."

Only participants aged 18 years or older who provided informed consent were eligible to proceed to the questions. Importantly, respondents were excluded if they declined participation or did not complete the questionnaire. In addition, they did not receive any form of credit or remuneration, and all data were collected anonymously and maintained in strict confidentiality.

### Minimal sample size calculation

For our study, a minimal sample size of 128 participants was calculated to ensure adequate statistical power, based on the formula proposed by Fritz and MacKinnon. Parameters for this calculation included a small to medium effect size (f = 0.26), L = 7.85 for an alpha error of 5%, a statistical power of 80% and 10 variables that were included in the final model. In the mediation analysis, CCA served as the predictor, PD as the outcome and hope as the mediator, with seven covariates: age, gender, marital status, education level, living arrangement, country of residence and history of mental illness. Eventually, our final sample of 2,844 participants far exceeded this requirement, providing adequate statistical power for the planned analysis.

Importantly, we particularly selected this effect size as a conservative assumption in line with methodological recommendations for mediation models, where indirect effects are usually smaller than direct ones (Fritz and MacKinnon, 2007; Fairchild et al., 2009; Preacher and Kelley, 2011). Indeed, we aimed with this strategy to prevent underestimating the required sample size, as much as possible, while still being consistent with the established standards for mediation analysis (Fritz and MacKinnon, 2007; Fairchild et al., 2009; Preacher and Kelley, 2011).

### Questionnaire and variables

As previously mentioned, our questionnaire was completely anonymous and was given in Arabic, since it is the native language of the participating countries. It was divided into three parts that required about 20 min to be filled. In the first section, the aims and objectives were clearly outlined, and the voluntary and confidential nature of participation was presented. Respondents were then required to provide informed consent electronically by choosing the option "I consent to participate in this study" before accessing the survey.

Sociodemographic information was gathered in the second section, where respondents were asked to state their age, marital status, gender, education level, country of residence, living arrangement as well as any history of mental illness.

Finally, the third section assessed the study's main variables using validated Arabic instruments: the *Climate Change Anxiety Scale (CCAS)*, the *PHS* and the *Patient Health Questionnaire-4 (PHQ-4)*.

#### Climate change anxiety scale

CCA was measured using an Arabic adaptation of the CCAS developed by Clayton and Karazsia (Fekih-Romdhane et al., 2024). The self-report instrument containing 13 items assesses affective and rational reactions to climate change using a five-point Likert scale from 1 ("never") to 5 ("almost always"). The main scales have two subdivisions: cognitive-emotional impairment (e.g., "Thinking about climate change makes it difficult for me to concentrate") and functional impairment (e.g., "My concerns about climate change interfere with my ability to get work or school assignments done"). Higher total scores indicate greater levels of climate-related anxiety. This version was validated among Arabic-speaking adults and showed exceptional internal consistency (Cronbach's $\alpha = 0.96$) and stable and strong two-factor structure (Fekih-Romdhane et al., 2024).

#### Perceived hope scale

Next, to measure hope, we used the Arabic version of the PHS, which consists of a six-item instrument designed to measure perceived agency and goal-directed pathways. The rating for each item is done on a five-point Likert scale from 1 ("strongly disagree") to 5 ("strongly agree"), with higher scores reflecting greater perceived hope (Fekih-Romdhane et al., 2025). The PHS assesses both goal-oriented thinking (e.g., "Even when I face difficulties, I can find ways to reach my goals") and motivational energy (e.g., "I am confident that I can find solutions to most of my problems"). It was validated in Arabic with excellent psychometric properties (Cronbach's alpha = 0.90). It is important to note that we chose the PHS rather than any other eco-hope or climate-specific measure to measure general hope as a psychological coping resource to examine its relationship with PD in the context of CCA (Fekih-Romdhane et al., 2025).

#### Patient health questionnaire-4

The Arabic version of the PHQ-4 scale was used to measure PD. This tool consists mainly of four items that are further subdivided into two subscales: one to measure anxiety and the other one to measure depression, each including two items. Participants rated each question on a four-point scale from 0 ("not at all") to 3 ("nearly every day") with a total score varying between 0 and 12. Scores of 3–5 reflect mild distress, 6–8 moderate, and 9–12 severe distress. The Arabic version demonstrated excellent internal consistency (Cronbach's $\alpha = 0.86$) and factorial validity, confirming a stable two-factor model (anxiety and depression) and measurement invariance across gender (Obeid et al., 2024).

### Statistical analysis

The SPSS v.27 software was used for the statistical analysis. The PD scores were normally distributed as shown by their skewness and kurtosis values between ±1. The Student t test was used to compare a continuous variable and a dichotomous variable, the ANOVA test to compare three or more means, and Pearson's test to correlate two continuous variables. The mediation analysis was performed using PROCESS MACRO (a SPSS add-on) v.4.2 Model 4. Four pathways

**Table 1.** Sociodemographic and other characteristics of the sample (*N* = 2,844)

| Variable | *N* (%) |
|---|---|
| **Gender** | |
| Male | 632 (22.2%) |
| Female | 2,212 (77.8%) |
| **Marital status** | |
| Single, divorced or widowed | 2,491 (87.6%) |
| Married | 353 (12.4%) |
| **Education** | |
| School level | 357 (12.6%) |
| University level | 2,487 (87.4%) |
| **Country** | |
| Jordan | 528 (18.6%) |
| Egypt | 1,418 (49.9%) |
| Palestine | 529 (18.6%) |
| Lebanon | 369 (13.0%) |
| **Living arrangement** | |
| Alone | 69 (2.4%) |
| With parents/partner | 2,730 (96.0%) |
| With friends | 45 (1.6%) |
| **Personal history of mental illness** | |
| No | 2,717 (95.5%) |
| Yes | 127 (4.5%) |
| | **Mean ± SD** |
| Age (years) | 22.46 ± 6.17 |
| Hope | 17.31 ± 7.20 |
| Climate change anxiety | 25.42 ± 10.07 |
| Psychological distress | 5.15 ± 3.41 |

were computed: Pathway A from climate change anxiety to hope, Pathway B from hope to PD, and Pathways C and C′ reflecting the total and direct associations of CCA with PD. Covariates entered in the model were those that showed a *p < 0.25* in the bivariate analysis. *P < 0.05* was considered statistically significant.

## Results

In total, 2,844 participants completed the questionnaire, with 77.8% females and a mean age of 22.46 years. The full description of the sample is in Table 1.

### Bivariate analysis of factors associated with depression and anxiety scores

A higher mean psychological distress score was found in female participants vs. males, in single, divorced or widowed individuals vs. married, in those with a school vs. university education level, in those living in Jordan vs. other countries and in participants who have a personal history of mental illness (Table 2). Furthermore, lower levels of hope and older age were associated with less

psychological distress, whereas higher CCA was associated with more PD (Table 3).

### Analysis of mediation

The mediation analysis taking the psychological distress scores as the dependent variable was adjusted over the following covariates: countries, sex, marital status, education level, age and personal history of mental illness. Hope partially mediated the association between CCA and PD (indirect effect: Beta = 0.003; Boot SE = 0.001; Boot CI 0.001; 0.005). Higher CCA was significantly associated with less hope, which in turn was significantly associated with lower PD. Additionally, higher CCA was significantly associated with more PD (Figure 1).

## Discussion

The present study investigated the mediating role of hope in the association between CCA and PD among a large multinational sample from four Arab countries. In line with our study objectives, the findings revealed that hope partially and modestly mediated this relationship, indicating that individuals experiencing greater levels of CCA tended to report lower levels of hope, which in turn was associated with higher PD. However, CCA remained directly associated with PD even after accounting for hope, suggesting that while hope contributes to explaining this association, it does not fully attenuate the psychological impact of climate-related concerns. Importantly, in this study, hope was examined as a general psychological construct that may be associated with PD in the context of CCA.

### Direct association between climate change anxiety and psychological distress

The significant direct link between CCA and PD observed in our study aligns with many previous studies associating CCA with adverse psychological outcomes. In fact, higher levels of climate anxiety were associated with greater symptoms of depression and anxiety, indicating a close relationship between emotional strain and chronic preoccupation with environmental and ecological destruction (Fritze et al., 2008; Sangervo et al., 2022; Ballew et al., 2024). One way to understand this link is through anticipatory anxiety. When people are repeatedly confronted with uncertain and seemingly uncontrollable future threats, emotional tension and persistent worry tend to increase. In line with this, Vercammen et al. showed that climate-related distress co-occurred with feelings of limited control, frustration over perceived inaction, guilt or shame about personal contributions, and disproportionate worry about someone's future, even in the absence of direct impacts (Vercammen et al., 2023).

Furthermore, repeated exposure to information about ecological crisis or lack of action from governments has the potential to alter, over time, how people view their own capacity to respond, reinforcing feelings of inefficacy and maybe futility with respect to those threats (Pihkala, 2020a). Indeed, several studies have shown that individuals who feel climate change is an immediate or ethical dilemma/threat often experience mental health implications, including concentration difficulties and sense of helplessness or lack of agency (Cianconi et al., 2020; Schwartz et al., 2023). Particularly, Cianconi et al. found that people who directly and/or indirectly were faced with climate change-related stressors, also

**Table 2.** Bivariate analysis of factors associated with psychological distress

| Variable | Mean ± SD | t / F | df / df1, df2 | P | Effect size |
|---|---|---|---|---|---|
| **Gender** | | −7.16 | 2,842 | **<0.001** | 0.323 |
| Male | 4.30 ± 3.28 | | | | |
| Female | 5.39 ± 3.41 | | | | |
| **Marital status** | | 3.45 | 2,842 | **<0.001** | 0.196 |
| Single, divorced or widowed | 5.23 ± 3.44 | | | | |
| Married | 4.56 ± 3.10 | | | | |
| **Education** | | 7.95 | 2,842 | **<0.001** | 0.450 |
| School level | 6.48 ± 3.69 | | | | |
| University level | 4.96 ± 3.32 | | | | |
| **Country** | | 23.52 | 3, 2,840 | **<0.001** | 0.024 |
| Jordan | 6.22 ± 3.40 | | | | |
| Egypt | 4.92 ± 3.36 | | | | |
| Palestine | 5.10 ± 3.38 | | | | |
| Lebanon | 4.58 ± 3.35 | | | | |
| **Living arrangement** | | 0.90 | 2, 2,841 | *0.408* | 0.001 |
| Alone | 5.61 ± 3.71 | | | | |
| With parents/partner | 5.14 ± 3.40 | | | | |
| With friends | 4.78 ± 3.24 | | | | |
| **Personal history of mental illness** | | −6.50 | 2,842 | **<0.001** | 0.642 |
| No | 5.05 ± 3.36 | | | | |
| Yes | 7.22 ± 3.69 | | | | |

Bold numbers indicate significant *p* value.

**Table 3.** Pearson correlation matrix

| | 1 | 2 | 3 |
|---|---|---|---|
| 1. Psychological distress | 1 | | |
| 2. Hope | −0.16*** | 1 | |
| 3. Climate change anxiety | 0.26*** | −0.06*** | 1 |
| 4. Age | −0.08*** | 0.03 | 0.06** |

**p < 0.01*; ****p < 0.001*.

experienced emotional and psychological outcomes, supporting the link between CCA and PD observed in our study, especially in Arab contexts, where climate change intersects with institutional and socioeconomic challenges. (Cianconi et al., 2020).

### Mediational role of hope

The mediation analysis in our study highlights hope as a partial and modest emotional pathway linking CCA to PD. According to Snyder's Hope theory, hope involves both goal-oriented thinking

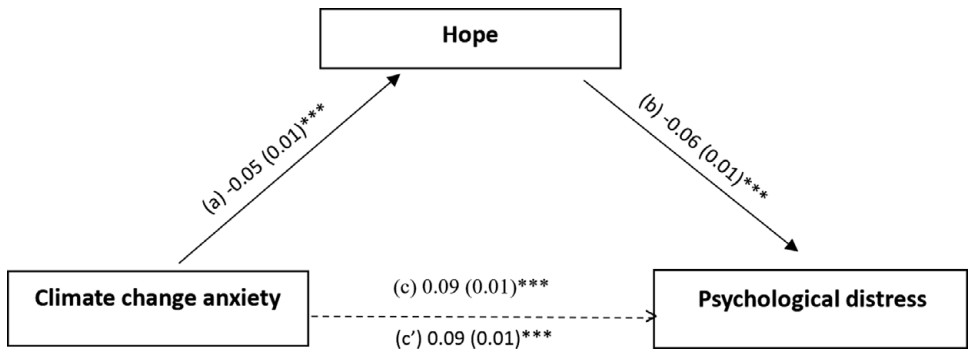

**Figure 1.** The mediation model taking hope as a mediator between climate change anxiety and psychological distress. (a) relation between climate change anxiety and hope ($R^2 = 0.040$); (b) relation between hope and psychological distress ($R^2 = 0.162$); (c) total effect of climate change anxiety on psychological distress ($R^2 = 0.144$); (c') direct effect of climate change anxiety and psychological distress. The numbers represent regression coefficients and their standard errors. ***p < 0.001*.

and the belief in one's ability to find strategies in order to achieve desired outcomes (Snyder, 2000). Moreover, individuals who experience CCA may feel a lack of control over future events, leading to a reduction in their sense of agency and personal efficacy. This results in giving up hope and feeling powerless (Siqueiros-García et al., 2022). This reduction in hope weakens the psychological capacity to cope with uncertainty, transforming concern about climate change from an adaptive motivator into a source of distress. Importantly, these processes were not directly measured in our study; however, prior research indicates that lower hope is associated with greater psychological vulnerability. Earlier studies pointed out that the most optimistic people with regard to environmental dangers are the ones to not only have the best psychological adjustment but also the most positive outlooks, whereas lower hope has been linked to elevated anxiety and depressive symptoms (Ojala, 2023; Yılmaz et al., 2025; Esbit et al., 2025). Therefore, it is important to note that our findings do not suggest that hope eliminates distress related to climate change, but rather reduced hope may represent one of many other mechanisms through which CCA is associated with poorer mental health and more PD.

Furthermore, Ojala indicates that hope is a factor that might enable people to balance out their negative feelings with positive engagement, thus being able to find meaning in the situation and regulate emotions when confronted with major and complex problems (Ojala, 2023). On the other hand, the decrease of hope leads to escalation of feelings of powerlessness and frustration, thus reinforcing maladaptive coping and withdrawal from collective actions (Vercammen et al., 2023). In line with this literature, our current findings suggest that the connection between climate anxiety and PD is not merely through the latter's emotional burden, but also through its association with lower hope. This is important in the Arab world, where the combination of very high vulnerability to climate change, alongside inadequate adaptation, can deprive people of their feelings of control and of their future outlook (Carnegie Endowment for International Peace, 2023.). In such an environment, the fluctuations in hope may be related to the emotional reactions of the people to the natural disasters; hence, the psychological suffering that is already associated with climate anxiety can be increased further.

It is important to note that although statistically significant, the indirect effect of hope was modest in magnitude, suggesting that hope represents one of several psychological factors involved in the association between CCA and PD. This pattern is consistent with the mathematical and conceptual properties of mediation models, in which indirect effects represent the product of multiple component paths and therefore tend to be smaller than direct effects (Fritz and MacKinnon, 2007; Preacher and Kelley, 2011). Effect-size indices for mediation are designed to quantify the proportion of variance in the outcome that is attributable specifically to the indirect pathway, which often seems to be small, particularly in large population-based samples, yet may still be statistically reliable and informative (Fritz and MacKinnon, 2007; Fairchild et al., 2009; Preacher and Kelley, 2011). Thus, a small effect size as observed in our study can still have practical relevance. In fact, even a modest contribution of hope may partially shape emotional responses at a population level, especially in Arab countries where environmental and socioeconomic challenges reduce people's ability to maintain a hopeful perspective. As a consequence, even small enhancements in hope via interventions or educational programs may lead to small benefits for psychological well-being and improvements in mental health, especially if combined with other supportive strategies.

Moreover, the persistence of a significant direct association between CCA and PD after accounting for hope suggests that psychological processes other than hope, like guilt or helplessness, that were not measured in our study may also be involved in this relationship (Pihkala, 2020b). As a consequence, the role of hope should be interpreted not as a dominant or central mechanism but as one of many other potential psychological resources that partially explain the association between CCA and PD. Thus, studying hope in these Arab countries matters not because it fully protects people from PD related to climate change but because persistent environmental challenges, coupled with socioeconomic difficulties, can reduce people's ability to feel positive about the future (Lawrance et al., 2022; Adom, 2024). Therefore, here hope may play a small role in shaping emotional responses to CCA, while external social and environmental challenges continue to have a stronger influence on PD.

## Implications and future directions

The present findings contribute to the existing literature by clarifying that the association between CCA and PD is only partially explained by hope. The persistence of a significant direct effect after accounting for hope indicated that climate-related PD potentially involves several mechanisms, including factors not examined in this present study. Therefore, the modest role of hope as a mediator should be interpreted as partial and supportive rather than fully explanatory. This suggests that multiple other psychological, social and even contextual factors are also likely involved in CCA (Pihkala, 2020b). In Arab contexts, studying hope is important not because it dominates psychological responses, but because it represents one of several measurable psychological factors, particularly in regions facing persistent socioeconomic challenges and limited institutional responses to climate change (Mahmood et al., 2025).

On the other hand, the mediational pathway alone highlights the potential value of integrating mental health support into climate change adaptation and public health strategies in the Arab countries. Although hope played a modest role in mediating the association between CCA and PD, our results suggest that interventions aiming to reduce climate-related distress could include strategies to foster and enhance hope, alongside coping strategies and general psychological support. The Arab region, being one of the most climate-vulnerable in the world and at the same time having limited adaptation and response strategies, requires coordinated regional action. Therefore, the alliance of healthcare/mental health professionals, educators, environmental nongovernmental organizations and governmental institutions could facilitate the development of culturally adapted interventions that promote hope and civic engagement. Besides, training of community counselors and teachers to identify signs of climate-related distress and to promote constructive coping may be used as a way to strengthen the outreach and durability of such initiatives.

Finally, our results reveal the need for longitudinal and intervention-based studies that explore how hope interacts with other psychological, social, cultural and structural factors over time and to determine whether changes in hope meaningfully alter psychological responses to CCA, particularly in regions facing numerous environmental and socioeconomic stressors.

## Limitations

Some limitations should be considered when interpreting the present findings. First, the use of convenience and snowball sampling approaches may have introduced selection bias and limited the generalizability and representativeness of the sample. The sample was predominantly female, young, educated and with a higher proportion of Egyptian participants, which could restrict the generalizability to the wider Arab population. While this sampling approach may limit, to a certain point, the external validity, it is less likely to compromise the internal associations and significance levels examined in the mediation analysis, which were adjusted for relevant covariates. As a result, the findings of our study should be interpreted as specific to this sample and cannot be considered representative of the general population.

Second, the cross-sectional design precludes causal inference, as the relationships between CCA, hope and PD were measured at a single point in time. As a result, the indirect effects in our mediation analysis reflect statistical associations rather than causal or temporal mechanisms. Therefore, the results should be interpreted as exploratory and hypothesis-generating.

Third, reliance on self-reported data may have led to response bias or inaccuracies due to social desirability and recall effects. Moreover, other psychological factors such as resilience or personality traits that were not evaluated in our study may have influenced the observed associations.

Furthermore, the use of tools that measure general hope and PD rather than climate-specific ones may have limited the ability to capture processes specific to CCA. Hence, future studies employing longitudinal or experimental designs using more representative sampling, climate-change-specific measure of hope and more diverse recruitment strategies are warranted to validate and extend these findings.

## Conclusion

This multinational study examined whether hope partially mediates the link between CCA and PD across Arab populations. The findings indicate a robust association between climate anxiety and distress, together with a partial mediating role of hope. That is, greater climate-related anxiety is tied to lower hope, and lower hope is, in turn, related to higher distress, while a direct association between climate anxiety and distress remains. These results suggest that hope represents one psychological factor that may influence how climate concern relates to PD. More culturally grounded longitudinal and experimental studies are required to determine how enhancing hope may reduce the mental health burden of climate change and to inform public mental health and climate adaptation strategies in the region.

**Open peer review.** To view the open peer review materials for this article, please visit http://doi.org/10.1017/gmh.2026.10191.

**Data availability statement.** The datasets produced and/or analyzed during this study are not publicly accessible but can be obtained from the corresponding author upon reasonable request.

**Acknowledgements.** K.J would like to acknowledge support through the ICTP-Arab Fund Associates Programme (2024-2026).

**Author contribution.** F.F.R. designed the study; T.B., F.F.R. and S.H. drafted the manuscript; S.H. carried out the analysis and interpreted the results; H.A.M.S., M.H., K.J., A.Y.N., M.F., E.R., A.A., and I.N. collected the data; D.M., S.O. and M.B. reviewed the paper for intellectual content; all authors reviewed the final manuscript and gave their consent.

**Competing interests.** The authors have nothing to disclose.

**Ethics statement.** We were granted ethical approval for our study from the Ethics and Research Committee of the Lebanese International University (reference number: 2022RC-051-LIUSOP). Subsequently, we adhered to the ethical guidelines stated in the Declaration of Helsinki and conformed to the institution's research regulations. The participation was completely voluntary, and the data collection process started only after obtaining electronic consent from all the respondents.

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
