## [Reviewer Report]

The authors have done a good job addressing most of the reviewer comments. However, several concerns remain regarding the language used, the interpretation of the results, and the overall contribution of the study. I also note some minor concerns related to newly added content, including the impact statement and the insufficient use of appropriate peer-reviewed references. These issues are outlined in detail below.

1. The authors have done a good job softening their claims to avoid overstating the findings. However, there are still a few sections where the language is too strong and definitive. For example:

• “On the other hand, the decrease of hope leads to escalation of feelings of powerlessness and frustration, thus reinforcing maladaptive coping and withdrawal from collective actions (Vercammen et al., 2023).” I would suggest that “may lead” would be more appropriate here.

• “Climate change can influence mental health through three different mechanisms including direct exposure to extreme weather events, indirect social and economic consequences, and anticipatory awareness of environmental deterioration (Berry et al., 2010; Bourque & Cunsolo Willox, 2014).” I would suggest “at least three different mechanisms,” as there are likely many mechanisms and pathways.

2. I have remaining concerns about the limitations of the study. Although I appreciate the expanded limitations section, the authors state that “the findings of our study should be interpreted as specific to this sample and cannot be considered representative of the general population.” This is a substantial limitation, yet it is not meaningfully addressed in the Discussion section. Although the authors have contextualized their findings within Arab nations, the manuscript would benefit from a clearer explanation of why these results are still important despite the acknowledged lack of generalizability.

3. The impact statement needs to be refined. It is unclear how the results of this study support the claim that: “this mediating role of hope suggests that interventions aimed towards addressing climate-related distress may require approaches that extend beyond reducing anxiety alone.” Perhaps it may be more accurate to state that interventions “may be more effective” if they extend beyond reducing anxiety, rather than that such approaches are required. Additionally, it is unclear how screening for symptoms of anxiety and depression constitutes a mental health strategy, as opposed to an assessment tool. Lastly, the specific emphasis on “both young and adult populations” is confusing.

4. In the Discussion section focused on the mediational role of hope, the first paragraph seems a bit disconnected from the results of this study. For example, the authors state that “these processes were not directly measured in our study.” Thus, it is not clear to me why this content was included in the Discussion section, and I would recommend removing or substantially reducing this paragraph. The subsequent paragraphs that directly engage with the study’s findings are much more appropriate and strengthen the discussion.

5. Several claims throughout the manuscript are not adequately supported by peer-reviewed literature. Please add high-quality references to support your claims, including but not limited to the following:

• “While this sampling approach may limit to a certain point the external validity, it is less likely to compromise the internal associations and significance levels examined in the mediation analysis, which were adjusted for relevant covariates.”

• “In fact, even a modest contribution of hope may partially shape emotional responses at a population level, especially in Arab countries where environmental and socioeconomic challenges reduce people’s ability to maintain a hopeful perspective.”

• “This is important in the Arab world, where the combination of very high vulnerability to climate change, alongside inadequate adaptation, can deprive people of their feelings of control and future outlook (Carnegie Endowment for International Peace, 2023).” A peer-reviewed source would be more appropriate here.

• “These countries were selected in our study because they represent contexts of high vulnerability to climate change combined with political and socioeconomic stressors…” Several claims in this paragraph rely on grey literature or are insufficiently supported.

• “The significant direct link between climate change anxiety and psychological distress observed in our study aligns with many previous studies associating climate change anxiety with adverse psychological outcomes.” Please cite specific peer-reviewed studies to support this statement.